# Diversity and Abundance of Potential Vectors of Rift Valley Fever Virus in the North Region of Cameroon

**DOI:** 10.3390/insects11110814

**Published:** 2020-11-19

**Authors:** Poueme Namegni Rodrigue Simonet, Njan-Nloga Alexandre Michel, Wade Abel, Eisenbarth Albert, Groschup Martin Hermann, Stoek Franziska

**Affiliations:** 1National Veterinary Laboratory Cameroon (LANAVET), Garoua BP 503, Cameroon; abelwade@gmail.com; 2Department of Biological Sciences, The University of Ngaoundere, Ngaoundere BP 454, Cameroon; njanalexandre@yahoo.fr; 3Institute of Novel and Emerging Infectious Diseases, Friedrich-Loeffler-Institut, 17493 Greifswald, Insel Riems, Germany; albert.eisenbarth@bnitm.de (E.A.); martin.groschup@fli.de (G.M.H.); franziska.stoek@fli.de (S.F.)

**Keywords:** Rift Valley fever, vectors, diversity, abundance, North Region, Cameroon

## Abstract

**Simple Summary:**

Rift Valley fever (RVF) is a mosquito-borne disease caused by the Rift Valley fever virus (RVFV) transmitted by various genera of mosquitoes usually classified into primary vectors and secondary vectors. The former, belonging to the genus *Aedes,* are known for their ability to lay drought resistant eggs that can maintain the virus on dry soil for many years in geomorphic structures in the form of shallow depressions. After heavy rains, mosquitoes hatch from these eggs, some of which are infected and transmit the virus to neighboring animals. The secondary vectors, mainly mosquitoes of the genera *Culex, Anopheles*, and *Mansonia*, can colonize these sites, reproduce in abundance, and subsequently spread RVFV. Although the northern regions of Cameroon host more than half of the country’s cattle, sheep, and goat populations, there is a dearth of information on the occurrence and transmission of RVFV and its vectors. The very common transhumance of animals during periods of drought leads to contact between domestic and wild animals and creates opportunities for cross-transmission of the virus. It also increases the possibilities of exposure of herds to vectors, in particular at water points. In addition, rare heavy rainfall, flooding, and irrigation-based agricultural practices in these regions provide conditions for vector proliferation and increase the risk of the spread of vector-borne diseases, including RVF. Therefore, this study aimed to determine species diversity and spatial distribution of potential RVFV vectors in the North Region of Cameroon. The study revealed the presence of potential primary and secondary vectors of RVFV with an abundance and a diversity varying according to the ecological sites studied. This presence of potential vectors with their variable number per trap, per night, or per site may create areas of variable risk for disease transmission to susceptible hosts. Molecular analysis (PCR) tests for RVFV RNA research and viral isolation methods on these vectors to determine their role in the epidemiology and control of RVF cannot be overemphasized.

**Abstract:**

Rift Valley fever (RVF) is a major viral zoonosis transmitted by mosquitoes. The virus is endemic in most parts of sub-Saharan Africa and can affect humans, livestock, and wild ungulates. Knowledge of the biology of vectors of Rift Valley fever virus (RVFV) is essential for the establishment of effective control measures of the disease. The objective of this study was to determine the species diversity and relative abundance of potential RVFV vectors in the North Region of Cameroon. Adult mosquitoes were trapped during the wet and dry seasons from December 2017 to January 2019 with “EVS Light” traps with CO_2_ baits placed at selected sites. The captured mosquitoes were identified using dichotomous keys according to standard procedures. The abundance was calculated with regard to site, zone, and collection season. A total of 27,851 mosquitoes belonging to four genera (*Aedes, Anopheles, Mansonia*, and *Culex*) and comprising 31 species were caught (including 22 secondary vectors (98.05%) and nine primary vectors (1.94%). The total number of mosquitoes varied significantly depending on the locality (*p*-value < 0.001). The average number of mosquitoes collected per trap night was significantly higher in irrigated areas (*p*-value < 0.001), compared to urban and non-irrigated areas. The study revealed the presence of potential primary and secondary vectors of RVFV with varying abundance and diversity according to locality and ecological site in the North Region of Cameroon. The results showed that the genus *Mansonia* with the species *Ma. uniformis* and *Ma. africana* formed the dominant taxon (52.33%), followed by the genera *Culex* (45.04%) and *Anopheles* (2.61%). The need for molecular analysis (PCR) tests for RVFV RNA research and viral isolation methods on these vectors to determine their role in the epidemiology and control of RVF cannot be overemphasized.

## 1. Introduction

Rift Valley fever (RVF) is an infectious disease of many wild and domestic animal species [1,2] caused by an RNA virus belonging to the order *Bunyavirales*, family *Phenuiviridae,* genus *Phlebovirus* [3,4]. The virus is transmitted by mosquitoes of several genera, including *Aedes* and *Culex* [5,6] and it was first isolated in 1931 during an outbreak in Kenya [7]. Infection with the virus causes abortions and high mortality in young animals. In humans, RVFV infection may be followed by a non-specific flu-like syndrome, but encephalitis and ocular as well as hemorrhagic syndrome can also occur [4]. Animals are mainly infected through bites of infected mosquitoes, while humans are typically exposed when they come into direct contact with infected bodily fluids and tissues of infected animals. Transmission to humans via mosquito bites has been associated to a milder disease and asymptomatic infections [8].

Rift Valley fever virus (RVFV) is one of the hemorrhagic fever viruses found in Africa [9]. Although epidemics of the disease have been occurring repeatedly in sub-Saharan Africa, there is limited knowledge on the maintenance of the virus during inter-epidemic periods. The factors that contribute to the re-emergence of the disease in hotspot areas are poorly understood. In addition, there are knowledge gaps in the understanding of critical aspects of the ecology of potential vectors and influences of the vector–virus–host interactions in the epidemiology of RVFV [3,6,9,10,11].

RVFV vectors can be classified into two major groups, namely primary and secondary vectors. The primary vectors, according to the literature, are mosquitoes of the genus *Aedes* [5,12,13] that maintain RVFV by transovarially transmitting the virus to the next generation [14]. They usually lay these infected eggs in geomorphic structures in the form of shallow depressions without a stream, which can be covered with grass and are called dambos in East Africa [14,15].

The eggs of these primary vectors are resistant to desiccation and can therefore diapause in dry depressions for long periods of time, and during periods of high precipitation, infectious mosquitoes can hatch. This can cause the virus to be transmitted to nearby animals and humans when the vectors search for blood meals. Once primary transmission of the virus has taken place, secondary vectors belonging to other genera, such as *Culex, Anopheles*, and *Mansonia*, which invade flooded grounds for breeding, contribute to the amplification of the virus, consequently resulting in outbreaks due to their ubiquitous biting patterns [16,17,18].

Following the outbreak of RVF in Mauritania in 1987, entomological and veterinary studies, initiated in Senegal to contribute to the understanding of the epidemiology of this arbovirosis, led to isolation of the virus in different mosquito species (such as *Aedes vexans* and *Ae. ochraceus*), suggesting a possible enzootic cycle of RVFV transmission in West Africa [19]. Additionally, this virus has been isolated from *Culex poicilipes, Ae. vexans, Mansonia africana*, and *Ae. fowleri* in Barkedji [20]. In Kenya, the primary vectors *Ae. mcintoshi* and *Ae. ochraceus* have been reported to serve as reservoirs of the virus [5,12,21].

Although the northern regions of Cameroon host more than half of the country’s cattle, sheep, and goat populations, there is a dearth of information on the occurrence and transmission of RVFV. In addition, transhumance of animals during periods of drought is common. Transhumance results in contact between domestic and wild animals and creates opportunities for cross-transmission of the virus. It also increases the possibilities of herds to be exposed to vectors, especially at watering points. Furthermore, infrequent high rainfall, flooding, and irrigation-based agricultural practices in the region provide suitable conditions for the proliferation of vectors and increase the risk of spread of vector-borne diseases, including RVF. The communities of the North Region of Cameroon are composed of pastoralists (30%) and agro-pastoralists (65%), practicing predominantly the traditional systems of husbandry [22]. The region is an important area for the production of small ruminants and cattle in Cameroon where the socio-economic, political, cultural, and religious activities of the farmers depend entirely on livestock.

Despite reports of RVFV-specific antibodies in livestock from the North Region of Cameroon [23,24], no entomological study on the potential vectors that play an important role in the transmission and spread of the virus [5,25] has been carried out before. In-depth knowledge of the bioecological parameters of vectors is essential to delineate risk areas and predict outbreaks. Therefore, this study was carried out in order to determine species diversity and spatial distribution of potential RVFV vectors in the North Region of Cameroon.

## 2. Materials and Methods

### 2.1. Study Site and Design

The study was carried out from December 2017 to January 2019 in four localities (Lagdo, Pitoa, Bokle, Garoua) of the Benoue administrative division of the North Region of Cameroon (Figure 1), which is a Sudano-Sahelian zone (8°36 to 12°54 LN and 12°30 to 15°42 LE) of low to medium altitude areas (average altitude: 249 m). The North Region is characterized by short rainy seasons from mid-March to October, an annual rainfall range of 1200 to 1600 mm, and an average ambient temperature range of 21 to 36 °C. A high quantity of temporary pools are created during the rainy season. The region is also characterized by the presence of numerous hydrographic networks, including the large and long Benoue River and a hydroelectric dam in Lagdo. Average monthly rainfall and temperatures during the collection period did not differ significantly from those of the preceding five years (*p*-value > 0.05; calculated on the basis of weather data provided by the Ministry of Transport; Regional Weather Service of North Cameroon in Garoua).

The proximity of the region to neighboring countries, high frequency of livestock trade, and the presence of permanent water points and fertile pastures favor the introduction and spread of transboundary diseases. In fact, the region is regularly frequented by transhumant herders and their livestock (cattle, sheep, and goats) from Niger, Nigeria, Sudan, and Chad during the dry seasons to access good pastures and water points. The vegetation of the North Region is characterized by the dominance of grassy savannas, shrubs, and thorny drought-resistant vegetation.

Based on the farming activities of rural populations in the region, and taking into account certain environmental parameters, such as hydrography, the study area was subdivided into three distinct ecological entities:(i)A rural irrigated area (RIA) in Lagdo (1851.61 km^2^) and Pitoa (1324.80 km^2^) with humid conditions due to large bodies of water used for crop irrigation and hydropower (large ponds and dams over 400 m in diameter). Usually, domestic (small ruminants, cattle, dogs, cats, horses, poultry, donkeys) and wild animals (monkeys, warthogs, rodents, reptiles, and wild birds) are found along this area.(ii)A non-irrigated rural area (NIRA) in Bokle (200.04 km^2^) with drier conditions and smaller bodies of water for market gardening activities dependent on the season. The vector hosts encountered along this area are similar to those found in the RIA.(iii)An urban area (UA) of Garoua (59.60 km^2^), characterized by dense human settlement. As a capture site in Garoua, we chose the surroundings of the zoological garden in the city center, which houses small ponds, gutters, and water points arranged for watering several species of animals in captivity (antelopes, birds, monkeys, hyenas, lions, and reptiles).

### 2.2. Vector Collection and Identification

Adult mosquitoes were sampled using two “EVS Light” traps (BioQuip Products Inc., Rancho Dominguez, CA, USA) with CO_2_ as bait placed outside at sampling sites shortly before sunset and collected before sunrise the next day. Dry ice served as source of CO_2_. With this capture method mainly the epidemiologically relevant hematophagous females are attracted. The trap collections were carried out (two traps at a single night per site once a month, consecutively for 14 months) on the study sites in the Benoue division of the North Region (Figure 1). At each collection point, the two traps were installed about 100 m apart to ensure good diffusion of the CO_2_ from the traps in order to attract mosquitoes from various horizons.

The trapped mosquitoes were killed by freezing in portable freezers at −20 °C and transported to the laboratory where they were identified by using dichotomous keys [26,27,28].

### 2.3. Data Analysis

Data on the diversity and abundance of mosquito species were entered into spreadsheets (MS Excel, Microsoft Corporation, Redmond, WA, USA). The captured vectors were split into primary vectors (genus *Aedes*) and secondary vectors (genera *Anopheles, Culex*, and *Mansonia*) [29]. The average number of mosquitoes collected per trap night (M/T/N) of each species was calculated for each capture site by dividing the total number of captured mosquitoes by the number of nights of capture and the number of traps per capture. Differences in fixed effects (areas, localities, and seasons) in each group were evaluated for significance with Generalized Linear Model (GML) approach. Statistical analysis was performed using SPSS software (IBM Corp. Released 2011. IBM SPSS Statistics for Windows, Version 20.0, IBM Corporation, Armonk, NY, USA). The significance level was set at *p*-value < 0.05 throughout the study.

## 3. Results

### 3.1. Species Diversity of Mosquitoes Collected

A total of 27,851 mosquitoes belonging to the four genera *Mansonia* (51.32%), *Culex* (44.17%), *Anopheles* (2.56%), *Aedes* (1.95%) (Table 1), and comprising 31 species were collected in four sampling locations (Appendix A). Overall, 27 species were identified in Pitoa and Bokle, while 20 and 23 species were identified in Garoua and Lagdo, respectively.

The total number of collected mosquitoes (Table 2) varied significantly depending on the locality (*p*-value < 0.001), with the largest number of mosquitoes being trapped in Lagdo (n = 9277) with an average number of mosquitoes per trap and night of 22.48, and the lowest number of mosquitoes in Bokle (n = 4372) with an average of 14.19 mosquitoes per trap and night (Appendix A). The number of specimens collected in the dry season (n = 15,415) was higher than that collected in the rainy season (n = 12,436), but this difference was not significant (*p*-value > 0.05). The total number of mosquitoes collected was significantly higher in the RIA (n = 17,188), followed by the UA (n = 6291), and the NIRA (n = 4372) (*p*-value < 0.001).

### 3.2. Abundance and Distribution of Potential Primary and Secondary Vectors of RVFV

The diversity and abundance of potential RVFV vectors were assessed at the four sampling sites (Table 1, Appendix A).

#### 3.2.1. Potential Primary Vectors

In this study, 542 (1.95%) specimens of the genus *Aedes*, considered as primary RVFV vectors, belonging to nine species (*Ae. aegypti*, *Ae. albopictus*, *Ae. circumluteolus*, *Ae. dalzieli*, *Ae. fowleri*, *Ae. mcintoshi*, *Ae. mucidus*, *Ae. ochraceus*, *Ae. vittatus*) were collected at the four localities. *Aedes aegypti* (22.14%) was the most represented species in this group, followed by *Ae. fowleri* (17.34%). With 42 individuals captured, *Ae. mcintoshi* (7.74%) was found only in the locality of Bokle. *Aedes albopictus* (5.90%) and *Ae. mucidus* (3.69%) were the least represented species of *Aedes* mosquitoes in the different capture localities. The specific composition of captured vectors and abundances based on the risk factor variables studied are presented in Table 2. As seen in this table, although the greatest number of primary vectors were captured in the rainy season (n = 536), and in the RIA (n = 255), abundance did not vary significantly with agricultural practices (irrigation).

#### 3.2.2. Potential Secondary Vectors

A total of 27,309 (98.05%) specimens belonging to 22 mosquito species were collected during the entire study period (Table 1; Appendix A). The genus *Mansonia* (n = 14,293; 52.34%) was the dominant taxon at all sites, followed by the genera *Culex* (n = 12,302; 45.05%) and *Anopheles* (n = 714; 2.61%). Two species were identified within the genus *Mansonia*: *Ma. africana* and *Ma. uniformis*. These two species presented average numbers of 92.59 and 44.87 specimens per trap and night, respectively. Despite this variability in abundance, these two species presented a homogeneous distribution, and no significant difference (*p*-value > 0.05) was observed between their spatial or temporal distribution. Among the other mosquito species, 12,302 (45.05%) specimens of *Culex* belonging to nine species were collected. Within this taxon, *Cx. quinquefasciatus* (n = 5144; 41.81%) was the most abundant, followed by *Cx. antennatus* (n = 4032; 32.76%), with the former showing a high density in Garoua (M/T/N: 128.12). The other species of the genus were present in very small numbers or absent from other capture sites. As for primary vectors (genus *Aedes*), the average number of secondary vectors varied significantly depending on the localities (*p*-value < 0.001). Additionally, the average number varied with the characteristics of the collection sites (*p*-value < 0.001) (Table 2). The number of secondary vectors was highest in Lagdo (n = 9161), followed by Pitoa (n = 7772), Garoua (n = 6112), and Bokle (n = 4264). The abundance of secondary vectors was significantly higher (*p*-value < 0.001) in the RIA (n = 16,933) than in the UA (n = 6112) and the NIRA (n = 4264).

## 4. Discussion

Knowledge on the bioecology of the potential vectors of RVFV is a fundamental aspect for determining the habitats most at risk of RVF emergence. The role of each vector depends on its presence, density, spatial and temporal dynamics, and host-biting preferences.

This study reports the first entomological work on RVFV vectors in the North Region of Cameroon. It revealed the presence of four genera of mosquitoes (*Aedes, Culex, Anopheles*, and *Mansonia*), with some species that have already been implicated as RVFV reservoirs or vectors during previous epizootics and epidemics in Kenya, Mauritania, Somalia, and Tanzania [19,29,30,31]. The composition and abundance of potential RVFV vectors in the study areas varied depending on the locality and agricultural activities based on irrigation practices. This is likely to create variable risk points for exposure of livestock to mosquito bites and subsequent disease outbreaks.

During this study, the largest number of mosquitoes was obtained in Lagdo and Pitoa, which corresponded to the rural irrigated area (RIA) characterized by the presence of irrigated crops throughout the year. These results corroborate with those of several other studies carried out in similar areas, revealing that cropping systems based on irrigation constitute productive grounds for some mosquito species compared to non-irrigated rural areas (NIRA) [32,33].

There was no significant difference in the average of mosquitoes during the rainy season compared to the dry season. These results differ to those obtained by Samwel et al. [29] and to what is known about the ecology of RVFV vectors. This could be explained by the precise choice of vector collection sites during this study. In fact, the trapping points at Lagdo and Pitoa were located just around the irrigated rice fields of the localities. In addition, the Lagdo hydroelectric dam ensures total irrigation of rice fields and fish ponds throughout the dry season, resulting in a high concentration of vectors. In Garoua, the traps were placed around the zoological garden, which shelters small ponds, gutters, and water points arranged for the watering of the wild animals in captivity. These factors may favor a strong proliferation and density of vectors in these areas during the dry season [19,33]. In addition, it is also possible that during the rainy season, heavy torrential rains accompanied by strong winds in North Cameroon could have caused larval dilutions and dispersal of mosquitoes during the nights of capture. Furthermore, bad weather with turbid water disrupts the breathing of larvae. Zeller et al. [19] have demonstrated that, during the arid season, the drying up of rivers leads to the displacement of animals towards the remaining water points with a high vector prevalence. The findings suggest that permanent irrigation mechanisms in localities with low rainfall favor the maintenance and even the increase in vector density during dry periods. The abundance of vectors during the dry season increases the duration of livestock exposure to mosquito bites and thus the probability of infection and spread of the virus.

The *Aedes* mosquitoes identified in this study have already been implicated as vectors of RVFV during numerous epizootics like the epizootics of 2006/2007 in northern Kenya [31]; and the epizootics of West Africa (Senegal, Mauritania) [19,34]. RVFV has been isolated in several of these species, such as: *Ae. circumluteolus* [35], *Ae. dalzieli* [5], *Ae. fowleri* [20,36], *Ae. mcintoshi* [14,35], *Ae. mucidus*, *Ae. ochraceus* [19]. Experimental studies have shown that *Ae. aegypti* and *Ae. albopictus* can biologically and mechanically transmit the virus [37,38,39]. It is possible that the collection method during the night led to an underestimation of the abundance of mosquitoes of the genus *Aedes*, especially of *Ae. aegypti* and *Ae. albopictus* because of their particularly diurnal activity. In fact, we collected mosquitoes from dusk (one hour before sunset) to dawn (just before sunrise), but a field study carried out in Reunion showed that, in females of these species, the host-seeking activity occurs throughout the day, with a strong peak occurring two hours before sunset and a less pronounced peak in the early morning [40]. This aspect should therefore be taken into account when formulating strategic plans for the implementation of preventive actions with a targeted control of RVFV vectors.

Within the different areas studied, homogeneity in the distribution of potential primary vectors was observed. Given the known role as RVFV reservoirs and epizootic initiators of these primary vectors, their presence could present a risk of viral transmission.

In the present study, the significant variation in secondary vectors according to locality and irrigation area suggests the potential risk of RVFV transmission and amplification in certain locations. Compared to the NIRA (localities of Bokle and Garoua), the high abundance of secondary vectors obtained in the RIA (Lagdo and Pitoa) is probably due to the abundance of suitable breeding sites, characterized by the permanent presence of irrigation water all year round and an abundant plant cover, which offers very good conditions for the emergence, proliferation, and resting of these mosquitoes [33,41,42]. As previously described, an increased risk of transmission may be associated with flood-prone areas such as Lagdo [29,32]. Due to their amplifying role, the abundant secondary vectors in Pitoa and Lagdo may be involved in accelerated virus transmission between animals and humans in the vicinity of our collection sites during an RVF outbreak, considering the maximum movement distance of mosquitoes, which is up to 620 m [20].

Although the results here suggest a strong correlation between the distribution of the vectors and the irrigated areas, risk areas could expand considerably depending on other factors, such as the presence of domestic vertebrate hosts (small ruminants, cattle, dogs, cats, horses, donkeys, poultry) and wildlife (monkeys, warthogs, rodents, reptiles, etc.) in these zones [43].

Poueme et al. [23] reported the presence of RVFV-specific antibodies in Cameroonian livestock and higher seroprevalence rates in Pitoa and Lagdo due to the present risk factors in these areas. The previous study of Rissmann et al. [24] also confirmed the circulation of RVFV in Cameroon. It is therefore necessary to set up a surveillance system in North Cameroon, to anticipate and respond to outbreaks of the disease.

This first study on the presence and diversity of the potential vectors of RVFV revealed that the genus *Mansonia*, with two species *Ma. africana* and *Ma. uniformis*, was most abundant at all collection localities. These mosquito species have been incriminated in epizootics and epidemics of RVF, notably in Kenya [35,42], Senegal [20], and the Central African Republic [44]. This high abundance of *Mansonia* spp. was probably due to adaptation of the species and the favorable biotope in the area. In fact, *Ma. africana* and *Ma. uniformis* preferably reproduce in ponds, irrigated gaps, and permanent waters with aquatic plants, living submerged due to an adapted siphon to obtain oxygen from the aerenchyma of those aquatic plants. These conditions have been observed in the various localities where dams or irrigation agricultural systems are encountered [28,32,45].

The findings also revealed a strong presence of mosquito species of the genus *Culex*, with predominant species such as *Cx. quinquefasciatus* and *Cx. antennatus*, which may also play an important role as secondary vectors in this semi-arid ecological site. The strong presence is enhanced by the capacity of their larvae to colonize urban and rural areas and to develop in various aquatic habitats [46]. Members of the *Cx. quinquefasciatus* complex have the ability to spread RVFV [35], and they have also been associated with the transmission of viruses such as West Nile virus (WNV) [47], Saint Louis encephalitis virus (SLEV) [48], and Venezuelan equine encephalitis virus (VEEV) [49]. Natural RVFV infection of *Cx. antennatus* has been reported in Kenya [14] and Nigeria [50,51]. However, *Cx. poicilipes*, *Cx. univittatus*, and *Cx. bitaeniorhynchus,* which are known to play secondary roles in the transmission of RVFV, have been captured less in this study.

The present study showed that *Cx. quinquefasciatus* and *Cx. antennatus* along with *Ma. africana* and *Ma. uniformis* constitute the most abundant secondary vectors of RVFV in the North Region of Cameroon and may therefore be associated with the amplification of RVFV in the event of an epidemic or epizootic. Other potential secondary vectors such as *Anopheles* spp. were scarcely present in the study sites. However, among the *Anopheles* mosquitoes collected, the species *An. squamosus* was found, previously infected with RVFV and associated with the RVF epidemic in Kenya in 2006/2007 [35].

Finally, it is important to note that several other factors can additionally modify or influence the specific composition or abundance of vectors in a given locality; for example, the existence of tracks or transhumance routes with the installation of watering points for animals can influence the reproductive patterns of mosquitoes and promote vector diversity [52].

## 5. Conclusions

This study revealed the presence of potential primary and secondary vectors of RVFV with an abundance and a diversity varying according to the ecological sites studied. This is an important contribution to the epidemiology of RVF or other mosquito-borne diseases in the North Region of Cameroon. The localities of Lagdo and Pitoa had a higher and diversified vector density compared to Garoua and Bokle. Many identified vectors have been implicated elsewhere in the maintenance and transmission of RVFV. The results showed that the presence of potential vectors with their variable number per trap and night, or per site may create areas of variable risk for disease transmission to susceptible hosts. The study can serve as a guide for the formulation of strategic plans for the implementation of preventive actions with a targeted fight against RVFV vectors. Molecular analysis (PCR) tests for RVFV RNA research and viral isolation methods on these vectors to determine their role in the epidemiology and control of RVF cannot be overemphasized.

## Figures and Tables

**Figure 1 insects-11-00814-f001:**
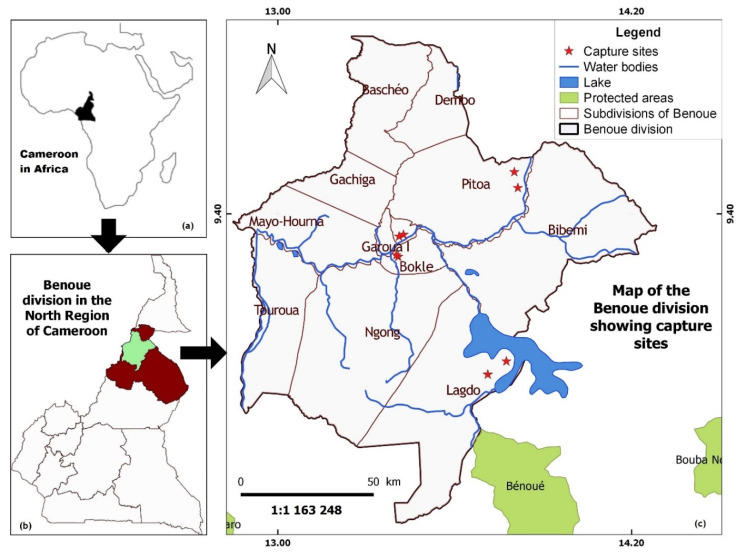
Map showing Cameroon in Africa, the study localities and sampling points in the North Region of Cameroon. (**a**) An insert of Africa map, showing Cameroon; (**b**) an insert of Cameroon, highlighting the Benoue division in the North Region; (**c**) an extract map, showing the study localities and sampling points in the Benoue division of the North Region.

**Table 1 insects-11-00814-t001:** Summary of mosquito genera captured: diversity, distribution, and abundance per locality.

Mosquito Genus	Localities (Trap Nights)								Total Number of Mosquitoes per Species	Mean M/T/N
	Bokle (10)		Garoua (13)		Lagdo (14)		Pitoa (15)			
	No.	M/T/N	No.	M/T/N	No.	M/T/N	No.	M/T/N		
*Aedes*	108	0.6	179	0.76	116	0.46	139	0.51	542	0.59
*Anopheles*	102	0.46	88	0.31	233	0.76	291	0.88	714	0.60
*Culex*	1129	5.13	4737	16.56	2029	6.59	4407	13.35	12,302	10.41
*Mansonia*	3033	50.55	1287	16.5	6899	82.13	3074	34.16	14,293	45.83
Number of mosquitoes per locality	4372	14.19	6291	8.53	9277	22.48	7911	12.23	27,851	14.36

No: number of mosquitoes caught; M/T/N: average number of mosquitoes collected per trap night (mosquitoes/trap/night).

**Table 2 insects-11-00814-t002:** Distribution of mosquitoes collected according to risk factors.

Risk Factor Variables			Number of Mosquitoes	Mean Number of Mosquitoes/Sampling	Standard Deviation	Levene Statistic	*p*-Value
Vectors collected	Season	Rainy Dry	12,436 15,415	26.02 25.11	19.69 18.17	16.86	0.41
	Localities	Bokle Garoua Lagdo Pitoa	4372 6291 9277 7911	17.77 29.54 27.45 26.82	12.16 16.70 19.62 22.00		<0.01 *
	Area	RIA NIRA UA	17,188 4372 6291	27.15 17.77 29.54	20.75 12.16 16.70		<0.01 *
	Total		27,851	25.50	18.84		
Primary vectors	Season	Rainy Dry	536 6	5.25 3.00	2.53 0.00	2.127	<0.01 *
	Localities	Bokle Garoua Lagdo Pitoa	108 179 116 139	6.35 4.97 6.11 4.34	2.98 2.46 2.40 2.09		0.034 *
	Area	RIA NIRA UA	255 108 179	5.00 6.35 4.97	2.35 2.98 2.46		0.12
	Total		542	5.21	2.52		
Secondary vectors	Season	Rainy Dry	11,900 15,409	31.65 25.18	18.50 18.15	33.17	<0.01 *
	Localities	Bokle Garoua Lagdo Pitoa	4264 6112 9161 7772	18.62 34.53 28.72 29.55	12.15 13.65 19.47 21.76		<0.01 *
	Area	RIA NIRA UA	16,933 4264 6112	29.09 18.62 34.53	20.52 12.15 13.65		<0.01 *
	Total		27,309	27.64	18.54		

*: significant *p*-value < 0.05; RIA: rural irrigated area; NIRA: non-irrigated rural area; UA: urban area.

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
