# Peer review of "Diversity and Abundance of Potential Vectors of Rift Valley Fever Virus in the North Region of Cameroon"

_insects, 2020, doi:10.3390/insects11110814_

Round 1

Reviewer 1 Report

The manuscript is improved, but there are a few minor issues that should be fixed.   

L39: replace is with “may be”

L56: Delete extra period.

L65 and throughout manuscript: Species names should be italicized, but the author of a species name is not italicized, for example, “Aedes ochraceus Theobald”. Use of author names for species names remains sporadic and inconsistent. As mentioned previously, I suggest including author names at the first mention of a species name. Similarly, do not begin sentences with abbreviated genus names, write out the full name (L177, L179, L288).

L83: specify that antibodies were detected in livestock.

L115: From where?

L119: Delete “town Garoua”

L142: How were data analyzed in R? L148 states that analyses were performed in SPSS.

L144: Suggest replacing Relative Abundance here and throughout with “average number collected per trap night” because relative abundance generally refers to the abundance of a species relative to other species, but here it seems to refer to the average number of a given species per trap night.

L242: Other than Ae. aegypti and Ae. albopictus, could the use of CDC traps have biased results for other species?

L206: What about vector competency? Host association may be a better term for trophic behavior here.

L241: Spacing issue before It

L262: Delete extra period.

References: Italicize species names and correct formatting errors throughout.

Author Response

Dear Reviewer
While saying thank you for your very relevant remarks and corrections which contribute to improving the quality of this manuscript, please find attached the word document which provides point by point answers to your corrections.
thank you so much

Reviewer 2 Report

I believe that the manuscript had improved and the additional information made it easier to understand. I still have few points I would to address and request to be careful with formatting the manuscript. I believe that the table and decimal separators are not in accordance.

Author Response

Dear Reviewer 2
While saying thank you for your very relevant remarks and corrections which contribute to improving the quality of this manuscript, please find attached the word document which provides point by point answers to your corrections.
thank you so much

Reviewer 3 Report

Line 100: It is still confusing – I think what you mean is

  1. i) A Rural Irrigated Area (RIA) in Lagdo (185161,1 ha) and Pitoa (132480,1 ha) with humid conditions from large bodies of water used for irrigating crops and hydroelectricity (large ponds and dams over 400m in diameter). Usually, domestic ……..
  2. ii) A Non-Irrigated Rural Area (NIRA) in Bokle (20004.2 ha) with drier conditions and smaller bodies of water for market gardening activities dependent on the season. The hosts …….

Line 134 – still not clear and no information on how many collection sites there were in each area – do you mean – (two traps at a single trapping site in each area, each month for 14 months)? The original questions – “Were traps put out on set days irrespective of weather conditions? Appears not to be the case – line 216. What potential hosts were there in the areas of the traps - was this a consideration in trap placement?” have not been clarified.

Line 118 – I see no mention of Aedes spp. in Table 1 or the Results. I cannot download the S1 file – not sure if that is the way the authors uploaded the data or the journal site - so it might be there. However, there should be some mention in the text of the number of samples that could not be identified because of damage and how this could have influenced results.

Line 205 – surely it is a relatively simple task to get weather data for the areas during the study and compare it with averages to see if it was a ‘normal’ or ‘exceptional’ year? If the study was carried out in a severe drought year or in a season characterized by severe rains and flooding it becomes of limited significance.  Readers need some sort of reassurance.

Line 136 – the paper is about potential vectors of RVF according to the title and the authors should then explain why they did not consider sex – do they consider males and females have equal vector competence? This needs to be addressed in the MS – why was sex not considered and how this could have influenced the value of the study.  

Author Response

Dear Reviewer 3
While saying thank you for your very relevant remarks and corrections which contribute to improving the quality of this manuscript, 

please, find attached the word document which provides point by point answers to your corrections.
thank you so much

Round 2

Reviewer 3 Report

The authors have addressed my queries.